# Effects of Heat Stress on Bovine Oocytes and Early Embryonic Development—An Update

**DOI:** 10.3390/cells11244073

**Published:** 2022-12-16

**Authors:** Klaudia Miętkiewska, Pawel Kordowitzki, Chandra S. Pareek

**Affiliations:** 1Department of Preclinical and Basic Sciences, Faculty of Biological and Veterinary Sciences, Nicolaus Copernicus University, 87-100 Torun, Poland; 2Division of Functional Genomics in Biological and Biomedical Research, Interdisciplinary Center for Modern Technologies, Nicolaus Copernicus University, 87-100 Torun, Poland

**Keywords:** cattle, heat stress, heat shock, oocyte, cumulus cell, granulosa cell, embryo, blastocyst, epigenetics, oxidative stress, mammary gland

## Abstract

Heat stress is a major threat to cattle reproduction today. It has been shown that the effect of high temperature not only has a negative effect on the hormonal balance, but also directly affects the quality of oocytes, disrupting the function of mitochondria, fragmenting their DNA and changing their maternal transcription. Studies suggest that the induction of HSP70 may reduce the apoptosis of granular layer cells caused by heat stress. It has been shown that the changes at the transcriptome level caused by heat stress are consistent with 46.4% of blastocyst development disorders. Cows from calves exposed to thermal stress in utero have a lower milk yield in their lifetime, exhibit immunological disorders, have a lower birth weight and display a shorter lifespan related to the expedited aging. In order to protect cow reproduction, the effects of heat stress at the intracellular and molecular levels should be tracked step by step, and the impacts of the dysregulation of thermal homeostasis (i.e., hyperthermy) should be taken into account.

## 1. Introduction

As homeothermic animals, cattle maintain a relatively constant body temperature; however, this can change depending on the temperature and humidity of the environment [1]. Global warming threatens cattle in many geographic areas, but it is European breeds that have reduced thermoregulatory capacity and experience severe effects from heat stress (HS) [2,3] The Holstein-Friesian breed is considered to be sensitive to high air temperature [4]. Currently, genetic selection and the intensification of milk production accelerate the metabolism of cows, which leads to the release of more metabolic heat. The metabolic rate of cows producing 31.6 kg of milk per day is 17% higher than those of cows producing 18.5 kg of milk per day [5]. Cows in conditions of high temperature and air humidity experience changes in physiological processes, meaning that they have to spend more energy on maintaining homeostasis [6,7]. When a cow’s body is exposed to high temperature, which is a strong stressor, the hypothalamic–pituitary–adrenocortical axis (HPA) and the sympathetic-adrenomedullary system (SAM) are activated, which is formed by the sympathetic autonomic and adrenal medulla. The HPA axis is neuroendocrine, which begins with the secretion of corticotropin-releasing hormone (CRH) and ends with the secretion of glucocorticoids [8].

The place that regulates and integrates the activity of these two systems is the hypothalamus, which, receiving signals from the external or internal environment, generates the appropriate response of the body. As a result of the activation of the SAM system, there is a greater secretion of catecholamines into the bloodstream and an increased concentration of adrenaline (A) and noradrenaline (NA), which—through adrenergic receptors—affect the functions of internal organs. This system is activated in the first moments of exposure to stress, and its purpose is to immediately cope with the stressful situation [9]. The HPA system, on the other hand, takes a while to activate, but its operation lasts for a longer time. Then, there is a cascade of hormonal changes, as a result of which, under the influence of adrenocorticotropic hormone (ACTH), cortisol, a glucocorticosteroid, is released from the adrenal cortex [8]. It has been shown that high temperature can increase cortisol secretion in dairy cattle [10]. However, there is conflicting information in the literature about cortisol concentrations in cattle when exposed to heat stress and cows under high-temperature conditions for a long time. As a result of prolonged exposure to high-temperature conditions, reduced cortisol concentrations may occur in cattle due to adrenal hyperplasia caused by chronic stress [11]. A prolonged secretion of cortisol has an effect on the immune system and on inflammatory processes, resulting in the induction of interleukin 6 (IL6) secretion [12]. Moreover, as a result of the SAM system action, interleukin 12 (IL12) is inhibited and interleukin 10 (IL10) secretion is stimulated, resulting in the inhibition of cell-mediated immunity [13]. This situation can lead to the immunosuppression and exposure of cattle to the development of numerous diseases [14]. Heat stress has been shown to have a direct effect on lipid and energy metabolism. In cows subjected to heat stress, a decrease in glucose and a decrease in non-esterified fatty acids (NEFA) in plasma were observed. However, an increase in urea and nitrogen was noted, which is probably the result of amino acid catabolism [15].

Increased plasma insulin levels have been observed in dairy cows during heat stress. This hormone is very important in the lipogenesis of adipose tissue. The observed reduced lipolysis under conditions of heat stress is probably a mechanism tending to reduce thermogenesis when exposed to thermal stress. In cows exposed to heat stress, there is also a decrease in blood plasma glucose concentration, as a result of which cows’ milk from cows exposed to heat stress contains less lactose [16].

Inappropriate cooling systems cause an increase in cows’ internal temperature during summer, leading to—among others—reduced fertility [17]. Heat stress indirectly reduces the fertility of cattle by inducing hormonal changes, affecting the uterine environment; as well as directly, causing a negative impact on oocytes and embryos [18]. Heat stress reduces the effectiveness of cattle artificial insemination (AI) and causes early embryonic death [3]. Exposing cows to an air temperature of 1.5–2 °C above normal body temperature may disturb the development of early embryos [19].

Thermal stress has a particularly negative effect on lactating dairy cows [4]. Another study showed a decrease in pregnancy rate from 71.2% to 45.7% due to HS [20]. In the United States, the financial loss from heat stress has been estimated at USD 1.2B per year [4,21,22].

Therefore, we aim to provide a better understanding of the mechanisms related to heat stress on bovine oocytes and early embryo development at the level of cellular changes, including transcriptomic and epigenomic changes, which hopefully, will help to develop effective strategies to support cow fertility.

## 2. Effect of Heat Stress on Bovine Oocytes

Although the period of oocyte maturation is relatively short compared to the entire life span of cows, exposure to HS during this period disrupts oocyte physiology, which has a negative impact on continued fertility [23]. It is well known that HS causes hormonal disruptions in cows, for instance, on the hypothalamic–pituitary axis. Previous research has provided strong evidence for the interference between HS and the secretion of gonadotropin-releasing hormone (GnRH), which leads to lower levels of circulating luteinizing hormone (LH), and follicle-stimulating hormone (FSH). As a consequence, this causes impaired growth of the ovarian follicles and disturbs the development of corpora lutea [24]. Studies have also shown that HS has a direct impact on the development and quality of bovine oocytes (Figure 1). It has been confirmed that during exposure to high temperatures, the oolemma of bovine oocytes consists of a greater amount of saturated fatty acids [25]. Changes in the composition of oocyte membranes have a negative impact on their further developmental competencies [26].

The production of heat shock proteins (HSPs), as a consequence of exposing oocytes to HS, is a favorable phenomenon for oocytes during the process of cryopreservation [27]. Furthermore, thermal stress may induce apoptosis in oocytes. It is worth noting that the protein HSP70 plays an important role in modulating apoptotic pathways [1]. Exposing cumulus–oocyte complexes (COCs) to high temperatures for 6–12 h reduces the developmental potential of oocytes during maturation and reduces the blastocyst yield [2]. Interestingly, it has been reported that exposure to an 8 h mild HS during bovine oocyte in vitro maturation (IVM) procedures does not adversely affect their developmental competence [24]. During the pre-antral period of ovarian follicle growth, intensive RNA synthesis in oocytes takes place. These follicles are extremely sensitive to environmental stimuli. Exposing cows to HS conditions during follicular growth or oocyte maturation leads to early oocyte aging and the impairment of their function [28]. Exposure to HS of bovine oocytes at the germinal vesicle (GV) stage could even lead to a weakening of nuclear and cytoplasmic maturation, and abnormal spindle formation [27]. Further on, it may disturb the function of oocytes’ mitochondria, resulting, among others, in DNA fragmentation, as a consequence of poor embryo quality [29]. It has been further reported that oocytes subjected to heat stress show a reduced migration of cortical granules and cristolysis of mitochondria [30]. As a result of exposure to thermal stress, many oocytes fail to progress to the metaphase II stage [22]. Moreover, GV-stage oocytes exposed to HS are described to have an altered maternal transcription and mRNA degradation [31]. More than half of maternal transcripts are silenced during oocyte maturation, and the synthesis of new proteins is dependent on the activation of stored transcripts. Exposure to HS during the latter-mentioned critical phase reduces the possibility of synthesizing new proteins [32]. It has been reported that the glutathione (GSH) content in bovine zygotes is much lower than in oocytes, suggesting that zygotes are more sensitive to HS than oocytes [33]. Interestingly, exposing cows to HS alters the relative amount of ATP synthase (ATP50) transcripts.

Heat stress has a negative effect on the oogenesis process, which impairs the reproductive capacity of cows. During the growth of ovarian follicles, the development potential of oocytes is shaped gradually. Disruption of this process caused by heat stress may lead to disorders of oocyte competence. The exposure of cattle to heat stress conditions leads to hormonal changes in their body and directly affects the structure of oocytes, thus, changing their function and reducing their quality. To keep the effects of HS on follicular physiology and subsequent fertility under control, it is important to carefully control cow reproduction, which should be taken into account to evaluate cow estrus and embryo distribution during hot periods [22].

## 3. Influence of Heat Stress on Granulosa Cells

Granulosa cells support the development of oocytes and provide an adequate microenvironment for maturation. Abnormalities in the functioning of granulosa cells may negatively affect the developmental competence of oocytes, and disrupt embryonic development and the establishment of pregnancy [34]. It has been shown that under the influence of high temperature, granulosa cells (GCs) lower the production of progesterone and estradiol. A reduced concentration of estradiol was accompanied by a weak expression of estrous signs, and a decrease in the level of progesterone impaired the fertility of cows [35]. The exposure of COCs to HS during the first half of meiotic maturation significantly reduces the blastocyst rate [36]. In particular, cumulus cells (CCs) have been shown to be sensitive to heat stress (Figure 2) [37]. The effects of HS are most detrimental during the first 10–12 h of maturation, during which the surrounding CCs are very closely related to the oocyte [38].

It is worth noting that the expansion of cumulus cells appears to be disturbed upon HS [39]. The exposure of CCs to high temperatures during in vitro maturation leads to induced DNA fragmentation in these cells [40]. Studies conducted by Latorraca et al. showed that one of the effects of high temperature on oocytes is autophagy [41].

Previous research has provided strong evidence that these key genes, which are involved in the response to HS, are activated [42]. It has been shown that the changes at the transcriptome level caused by HS are consistent with 46.4% of blastocyst development disorders [43]. The studies conducted by Rispoli et al. [18] showed that the relative amount of Caveolin 1 (CAV1) transcripts was 1.5 times lower in cumulus cells that were exposed to high temperatures during in vitro maturation. It was also observed that the exposure of oocytes to 41 °C resulted in a 2.5-fold reduction in the *Matrix Metalloproteinase-9 (MMP9)* transcript levels of CCs. In this regard, it is worth mentioning that appropriate levels of *MMP9* are associated with oocyte developmental competence, high pregnancy rates and effective embryonic implantation [44]. There was an increase in *Nitric Oxide Synthase 2 (NOS2)* transcripts in CCs exposed to high temperatures [45]. However, it has been shown that HS has no effects on the relative mRNA abundance of *Matrix Metallopeptidase-2 (MMP2)* [1]. Heat stress leads to oxidative stress and apoptosis through the intracellular accumulation of reactive oxygen species (ROS). Research suggests that the upregulation of *superoxide dismutase 2 (SOD)* and *catalase (CAT)* may regulate *forkhead box O3 (FOXO3)* and *kelch-like ECH-associated protein 1 (KEAP1)* in granulosa cells, thereby inhibiting ROS synthesis [17]. As a result of heat stress, transcriptional changes in GCs have also been observed, including an increase in the synthesis of HSP70, HSP90, HSP27 and HSP60 [42]. Studies suggest that the induction of HSP70 may reduce the apoptosis of GCs caused by heat stress [46]. However, other studies report that an increased level of HSP70 leads to a reduction in the aromatase protein in antral follicles, which reduces the fertility of cows [42]. Overall, HS leads to several cellular and physiological changes in the functioning of GCs, thus, worsening the fertility of dairy cattle [40].

Deterioration of the quality of the granular cell layer as a result of heat stress may indirectly affect the development of the ovarian follicle and disrupt the proper course of the fertilization process and further pregnancy.

## 4. Influence of Heat Stress on the Development of Bovine Embryos

During the first stages of bovine embryo cleavage (before the 8–16 cell stage), the genome is not yet active and the embryo is sensitive to many stressors, including high temperature [3]. Bovine embryos become resistant to heat shock at about the same time as the activation of the embryonic genome (this is the stage of embryo development when it contains between 8 and 16 cells) [47]. The process leading to the achievement of thermotolerance involves cellular mechanisms, including the production of HSP, as a result of which it is possible to induce cell resistance to further stressors [48]. Exposure of Holstein heifers to high temperatures 7 days after the onset of heat was shown to increase the number of abnormally developing embryos [49]. The negative influence of high temperature on the in vitro embryo production (IVP) process was also demonstrated. It was confirmed that the exposure of zygotes to the temperature of 40–41 °C resulted in a reduction in blastocyst rate [50]. On the other hand, in the case of 8-cell stage embryos, the effects of high temperature depended on the duration of heat stress and its severity [20].

As embryonic development progresses, embryos become more resistant to high temperatures. Reduced development was observed when embryos were exposed to a temperature of 41 °C on day 0 and day 2 of development, whereas there were no detectable effects when exposure occurred on days 4 and 6 [4,51]. When bovine morulae were confronted with a temperature of 40 °C for 8 h, the expression of genes encoding for *HSP family B member 11* (HSPB11) and *HSP family A (Hsp70) member 1A (HSPA1A)* was enhanced, in order to protect against the negative effects of high temperature, including the stabilization of ribosomal RNA [19,52]. HS also increased the amount of heat shock protein 90-alpha A1 (HSP90AA1) [44]. It has been proven that two-cell and four-cell embryos are capable of heat shock-induced activation of the transcriptional response before the main genome is activated [53]. The early embryo’s ability to respond to heat shock is inconclusive [54]. Scientific research gives different information regarding induced thermotolerance in two-cell embryos. Some sources say that in the situation of induced thermotolerance, the exposure of embryos to mild heat shock leads to cell immunity through HSP synthesis, while other studies indicate that two-cell embryos are unable to survive the induced thermotolerance [55]. Thermotolerance involves the transcriptional activation of *HSP* genes [56].

The effect of high temperature on early embryos leads to many cytoplasmic changes, including the reorganization of the cytoskeleton, precipitation of chromatin, migration of organelles to the center of blastomeres, and mitochondrial edema, which has a negative impact on the further separation of chromosomes and the proper development of embryos [4,53]. Maternal mitochondria are necessary to provide energy for initial cell division, as well as for calcium homeostasis [57]. It has been shown that mitochondrial dysfunction in the oocyte is inherited by the embryo and may affect the further development of the fetus and placenta. Certain mitochondria in cattle are degraded during cleavage, which may compromise embryo development, while the remaining mitochondria will be exposed to heat stress [23,52]. Among the pathways influencing the later development of the embryo and pregnancy is the WNT signaling pathway, in which exposure to high temperature was shown to result in an increase in *Axis Inhibition Protein 1* (AXIN1) and *Lymphoid Enhancer-Binding Factor 1 (LEF1)* expression, and a decrease in *Cullin-1 (CUL 1)* and the gene-encoding *PP2A (PPP2A)* [19]. DNA fragmentation characteristic of apoptosis was observed in the embryos exposed to HS [58]. Previous research has provided strong evidence that temperature influences the epigenome of embryos, which may have a negative impact on their development, and errors in methylation may be inherited in the next generations [55]. Mouse embryos exposed to high temperature had insufficiently *methylated paternal H19* and *Insulin-Like Growth Factor 2 Receptor (IGF-2R)* genes [4]. Heat stress may induce epigenetic disturbances, which might impact the expression of *HSP70*, which is considered a candidate heat stress biomarker [7]. The two-cell embryo is very sensitive to heat stress and may increase the transcription of *Hsp70 Member 1A (HSPA1A)*, *Heat Shock Protein beta-1 (HSPB1)*, and the synthesis of *HSP70*, when exposed to HS [17].

## 5. Thermal Stress and Its Influence on Pregnancy and Fetal Development

Heat stress has negative effects on pregnant cows and may in utero affect fetal development [52]. Especially during late pregnancy, namely during the dry period of dairy cows, HS may shorten gestation length by approximately 1.1%, compared to cows that are provided with a cooling system during warm seasons [53,59]. It has been shown that female fetuses exposed to thermal stress in utero give rise to cows with a lower milk yield, exhibit immunological disorders and have a shorter lifespan [2,60]. Primiparae generated from mothers that have been exposed to heat stress produce less milk in the first lactation [61]. The authors report that the exposure of cows to HS during the dry period has intergenerational effects that might be effective for up to two generations [62]. Fetal growth is also at risk due to placental insufficiency caused by hyperthermia. As a consequence, the transport of oxygen and nutrients between the mother and the fetus is disturbed [60]. It was also shown that the cotyledons of heat-stressed cows in the last 46 days of gestation had a larger surface area compared to those from non-stressed cows. However, thermal stress reduced the weight of the placenta [59]. A reduced amount of total DNA and RNA in the placenta has been shown in cows exposed to heat stress [5]. The effects of high temperature on the uterus could lead to epigenetic modifications in the fetus that affect newborn calves [63]. Furthermore, there are possible effects of HS on the fetal endocrine system, which result in hormonal changes in newborn calves, such as increased insulin levels in calves’ blood during their first weeks post-natal [21]. The colostrum consumption of calves exposed to HS in utero has been reported to be less effective, and the quality of colostrum itself is reduced (Figure 3) [64].

Newborn calves from heat-stressed cows have a higher rectal temperature compared to calves from mothers that were kept in a cooler environment in the last stages of pregnancy [61]. Understanding the mechanisms of heat stress in the course of pregnancy in cattle offers a chance to reduce economic losses, increase herd productivity and, above all, protect cows’ welfare [61].

## 6. Heat Stress and Mammary Gland Function and Milk Quality

The proper functioning of the mammary gland during the dry period and lactation is crucial for maintaining high milk yield and milk quality (Figure 3). In lactating dairy cows, reduced udder blood flow occurs as a result of reduced feed intake [5]. Heat stress can endanger the functioning of the mammary gland during the dry period when its cells proliferate. The proper course of this period is very important for obtaining high milk production in the upcoming lactations [65]. It has been shown that as a result of heat stress, cows use more energy for the synthesis of fatty acids, instead of the proliferation of mammary gland cells [5]. It is said that maternal heat stress may alter the epigenetic profiles of the developing fetus’ mammary gland, but this issue requires further research [5,63].

High air temperature leads to an increase in udder temperature, which results in a reduced expression of genes involved in the construction or biosynthesis of epithelial cells and an increased expression of genes responsible for protein repair [66]. Exposure of the mammary gland to high temperatures disrupts important cellular processes such as apoptosis and autophagy. This may disrupt the process of involution [67]. However, studies also show that as a result of exposure to heat stress, there is increased apoptosis in the primary cells of the mammary gland, which may result in a decrease in the total number of cells of the epithelium that builds it [5,65]

It has been shown that, as a result of heat stress, *HSP70* mRNA expression is induced in the epithelial cells of the mammary gland to protect against protein aggregation and degradation [5,66]. A change in the transcriptome of the mammary gland cells was also observed during heat stress, which indicates impaired mammary gland development during this time and abnormal cellular processes [68]. Heat stress on the fetus negatively affects the development of mammary gland cells in future heifers, leading to impaired mammary function in their future life [61]. During lactation, exposure to high temperatures causes glucose uptake by tissues, which reduces its availability in mammary gland cells [69]. Exposure to heat stress leads to downregulated metabolic pathways related to the metabolism of glucose and fatty acids [70].

As mentioned earlier, heat stress leads to a decrease in milk yield. Reduced milk yield can result from various mechanisms such as reduced feed intake and a disturbing number and secretory activity of mammary cells [5,65]. In addition to the effect of heat stress on milk yield, there are also changes in milk quality. During periods of heat, there is a decrease in the percentage of fat in the milk [71]. Then, there is also an increase in the proportion of long-chain fatty acids and a decrease in medium-chain and short-chain fatty acids [70]. It has been shown that thermal stress can also affect the level of protein in milk, usually causing a decrease in its level [71]. There is also an increase in somatic cells in milk from overheated cows [61,72]. Heat stress can permanently alter the development of the mammary gland and can be detrimental not only to milk production, but also to its quality.

## 7. How to Counteract the Development of Heat Stress?

Trying to prevent heat stress in dairy cows has become one of the most important paths to improving the quality of oocytes, embryos, and thus, the quality of female reproduction (Figure 4). A few ideas can be used and developed to decrease the impact of heat stress on this aspect. One of the most drastic effects of heat stress that can enormously impact oocytes’ and embryos’ quality are increased levels of reactive oxygen species (ROS) [73]. Heat stress increases ROS production in bovine oocytes [74] and reduces the number of ovarian follicles that can properly develop [75]. A few antioxidants might have a positive impact on this issue, e.g., retinol [76], cysteine [74] or melatonin [77]. The addition of melatonin to IVM media reduces the generation of ROS in maturing, and therefore, elevated oocytes’ quality. Retinol is another antioxidant that can be useful to improve the developmental competence of oocytes [78]. It has been suggested that Insulin-like growth factor 1 (IGF1) may support bovine embryos affected by HS (Figure 4) [2,74,79,80]. The in vitro treatment of embryos with IGF1 was shown to improve the embryo survival rate after transfer into heat-stressed recipients [81]. IGF1 also improves the resistance of day 4–6 embryos to heat shock and oxidative stress [2,81,82]; however, no effects of IGF1 on two-cell embryos were observed [79]. Other research indicates that the supplementation of IVM media with IGF1 decreases the negative effects of HS in oocytes generated from buffalos [83]. HSP70 is another protein that was examined in the conditions of heat stress to improve the IVM rate of bovine oocytes [2]. HSP70 is a chaperon that stabilizes unfolded or partially folded proteins; it prevents inappropriate inter- and intramolecular interactions [84]. The research that compared bovine IVM conditions, both with and without HSP70, showed that the presence of exogenous HSP70 may decrease the deleterious impact of high temperature on blastocyst yield (Figure 4) [85]. Bearing in mind that the ability of cows to maintain normothermia depends not only on metabolic heat production, but also on environmental conditions such as air temperature, air humidity, solar radiation, and others, the use of various methods of cooling cows seems to be a good solution for their protection against the effects of heat stress [86,87]. The architecture of barns and facilities that house dairy cows should be designed in such a way that provides cows with shade and a cooler temperature during hot seasons. In the US, the annual economic loss for cows kept in the sun was set at USD 1.5B, while placing the cows in the shade led to a 43% reduction in this value [88]. It is very important to create air movement to cool the surface of the animals’ skin; as reported previously, an air velocity of 0.2–0.9 m/s is sufficient for cows [89]. At the moment, the best solution for cooling cows under heat stress conditions seems to be direct evaporative cooling, which involves the simultaneous use of sprinklers and fans, allowing the moisturizing and cooling of cattle’s skin [86,90]. It is suggested that hormone therapies may reduce the effects of heat stress on cows. Striving to increase the level of progesterone in plasma gives a chance to compensate for its possible decreases as a result of thermal stress. However, studies show that embryo transfer is the best solution to protect the reproduction of cows living in hot conditions [35]. In recent decades, methods have been successively developed to combat the negative effects of heat stress. It is very important to understand the mechanism of this type of stress, in order to be aware of access to the possibilities of protecting the herd against its effects. To sum up, several ideas have been developed to reduce the effects of heat stress on bovine oocytes and embryos, and to find the best solutions to increase the quality of bovine reproduction.

Abbreviations: ACTH—adrenocorticotropic hormone; CRH—corticotropin-releasing hormone; GnRH–gonadotropin-releasing hormone; HSP–heat shock protein; IGF—insulin-like growth factor; IL-interleukin.

## 8. Conclusions

Heat stress has a negative effect on most reproductive functions of cows. The exposure of cattle to high temperatures leads to overheating of the reproductive organs, disrupting the process of oogenesis and embryogenesis. In addition, heat stress leads to hormonal changes, reducing estrus symptoms. The effects of heat stress on the body of cows are probably the result of the animals’ attempts to adapt to heat conditions by trying to minimize endogenous heat production. Such a situation leads to disturbances in the energy metabolism in oocytes and embryos, which fundamentally impacts their developmental competence. Finally, precisely recognizing the biological factors responsible for the heat stress-mediated induction of impairments in reproductive physiology and endocrinology in cattle and other mammalian species might enable the multifaceted estimation of genomic, epigenomic, transcriptomic and proteomic profiles in the embryos, fetuses and offspring created and multiplied by such novel assisted reproductive technologies and in vitro embryo production strategies as the ex vivo fertilization of oocytes, with the aid of either classic penetration by capacitated spermatozoa [91,92] or intra-cytoplasmic sperm injection [93,94] and somatic cell cloning [95,96]. The harmful effects of heat stress on cow reproduction are of key importance to the economic status of the farm, producing financial losses every year. Maintaining the proper fertility of cows in a situation where the temperature and air humidity exceed the values specified for cows is very important for maintaining the profitability of the farm. Every effort should be made to ensure that cows do not accumulate excessive body heat and that they can easily eliminate incremental heat loads. The literature states that the cellular mechanisms in oocytes caused by high temperatures are not fully explored. However, we believe that the evidence that some mechanisms occurring during heat stress are related to the increase in reactive oxygen species supports the use of antioxidants for the thermal protection of oocytes during the in vitro maturation process. We also consider it particularly important that IGF1 has an effect on the increased survival of embryos exposed to heat stress during the in vitro process. Undoubtedly, an interesting issue is the positive effect of HSP70 on the developmental competence of oocytes subjected to heat stress, and the resulting clear improvement in their quality during in vitro production. A very important aspect of improving the quality of oocytes and embryos in vivo is the cooling of cows during hot days. This is a very important issue to protect the health of dairy cattle and the developing pregnancy, and it contributes to improving the profitability of the farm. Most important is to recognize heat stress conditions as early as possible, in order to implement strategies to prevent their further development as soon as possible. Understanding the cellular and molecular changes that occur in oocytes, granulosa cells and embryos exposed to heat stress can help develop strategies to counter the effects of overheating.

## Figures and Tables

**Figure 1 cells-11-04073-f001:**
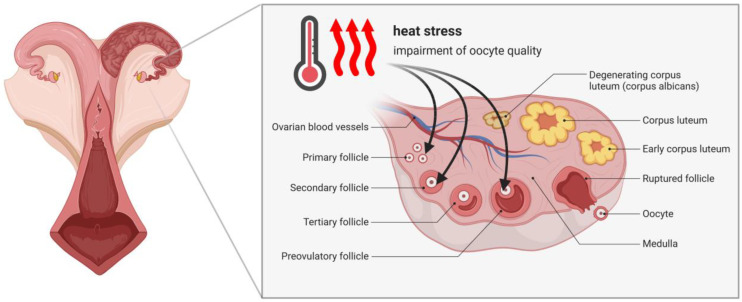
Scheme showing the impact of elevated environmental temperature (heat stress) on the impairment of oocytes at different stages of follicular development.

**Figure 2 cells-11-04073-f002:**
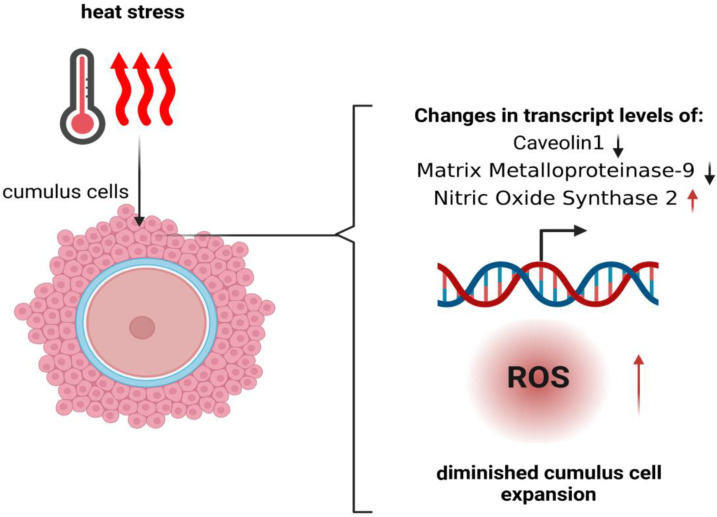
Scheme showing the impact of heat stress on cumulus cells on transcript levels, reactive oxygen (ROS) levels inside cumulus cells, and the expansion of cumulus cells, which is crucial during oocyte maturation.

**Figure 3 cells-11-04073-f003:**
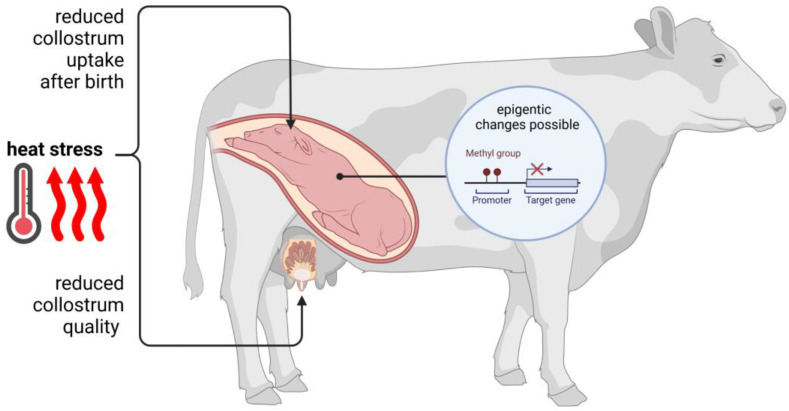
Scheme showing the impact of heat stress on developing fetus (calf) and on the mammary gland during bovine pregnancy.

**Figure 4 cells-11-04073-f004:**
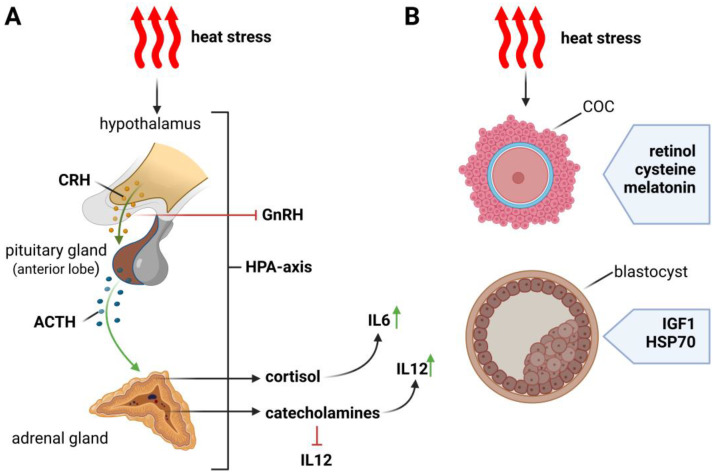
Scheme showing the impact of heat stress on the hypothalamus–pituitary–adrenal (HPA) axis (**A**), and how to counteract the effect of heat stress on the cumulus–oocyte complex (COC) and the blastocyst (**B**).

## Data Availability

Not applicable.

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
