# Peer review of "Effects of Heat Stress on Bovine Oocytes and Early Embryonic Development—An Update"

_cells, 2022, doi:10.3390/cells11244073_

Round 1

Reviewer 1 Report

The review of heat stress in bovine oocyte and early embryo development by the authors can not well explain the specific mechanism of heat stress. It is just a simple listing of some relevant influencing factors, and it is recommended to make a more in-depth summary and induction. The mechanism of heat stress on the development of bovine oocytes and early embryos was further explained. For example, add some flow charts of heat stress related mechanisms. This will enable more in-depth discovery.

Author Response

MS ID: cells-2073278

Dear Academic Editor,

Dear Reviewers,

Thank you for inviting us to respond to the very thoughtful and constructive reviewer comments. We greatly appreciate the Editor’s and Reviewers’time and believe our revised manuscript has become more well-rounded as a result.

We have incorporated all suggestions throughout the manuscript, which are highlighted in red in the attached file. Below is a point-by-point response to reviewers’ comments to clarify which edits were made.

We are happy to respond to additional requests if they arise.

Sincerely,

Chandra S. Pareek

Please note our following explanations:

Detailed answers to Reviewer 1:

REVIEWER: “The review of heat stress in bovine oocyte and early embryo development by the authors cannot well explain the specific mechanism of heat stress.”

Answer: Thank you for drawing attention to the importance of the mechanism of heat stress in dairy cattle. We decided to clarify this issue and added a new figure 4 and a more detailed explanation in the introduction of the article as follows:

When a cow's body is exposed to high temperature, which is a strong stressor, the hypothalamic-pituitary-adrenocortical axis (HPA) and the sympathetic-adrenomedullary system (SAM) are activated, which is formed by the sympathetic autonomic and adrenal medulla. The HPA axis is neuroendocrine, which begins with the secretion of corticotropin-releasing hormone (CRH) and ends with the secretion of glucocorticoids [Gunnar]. The place that regulates and integrates the activity of these two systems is the hypothalamus, which, receiving signals from the external or internal environment, generates the appropriate response of the body. As a result of the activation of the SAM system, there is a greater secretion of catecholamines into the bloodstream and an increased concentration of adrenaline (A) and noradrenaline (NA), which through adrenergic receptors affect the functions of internal organs. This system is activated in the first moments of exposure to stress, and its purpose is to immediately cope with the stressful situation [Valentino]. The HPA system, on the other hand, takes a while to start up, but its operation lasts for a longer time. Then there is a cascade of hormonal changes, as a result of which, under the influence of adrenocorticotropic hormone (ACTH), cortisol, a glucocorticosteroid, is released from the adrenal cortex [Gunnar]. It has been shown that high temperatures can increase cortisol secretion in dairy cattle [Kaufmann]. However, there is conflicting information in the literature about cortisol concentrations in cattle when exposed to heat stress and cows are under high temperature conditions for a long time. As a result of prolonged exposure to high-temperature conditions, reduced cortisol concentrations may occur in cattle due to adrenal hyperplasia caused by chronic stress [Ronchi B]. Prolonged secretion of cortisol has an effect on the immune system and on inflammatory processes, resulting in the induction of interleukin 6 (IL6) secretion [Van Gool]. Moreover, as a result of the SAM system action, interleukin 12 (IL12) is inhibited and interleukin 10 (IL10) secretion is stimulated, resulting in the inhibition of cell-mediated immunity [Elenkov]. This situation can lead to immunosuppression and exposure of cattle to the development of numerous diseases [Bagath]. Heat stress has been shown to have a direct effect on lipid and energy metabolism. In cows subjected to heat stress, a decrease in glucose and a decrease in non-esterified fatty acids (NEFA) in plasma was observed. However, an increase in urea and nitrogen was noted, which is probably the result of amino acid catabolism. Increased plasma insulin levels have been observed in dairy cows during heat stress. This hormone is very important in the lipogenesis of adipose tissue. The observed reduced lipolysis under conditions of heat stress is probably a mechanism tending to reduce thermogenesis when exposed to thermal stress. In cows exposed to heat stress, there is also a decrease in blood plasma glucose concentration, as a result of which cows' milk of cows exposed to heat stress contains less lactose.

REVIEWER: It is just a simple listing of some relevant influencing factors, and it is recommended to make a more in-depth summary and induction.

Answer:  We thank the Reviewer for the suggestion to enrich the introduction and summary of our article with more detailed knowledge and conclusions. We decided to include information on the mechanism of heat stress development in the introduction, as shown above, and the conclusions have been modified as follows:

Finally, precisely recognizing the biological factors responsible for heat stress-mediated induction of impairments in reproductive physiology and endocrinology in cattle and other mammalian species might bring about the multifaceted estimation of genomic, epigenomic, transcriptomic and proteomic profiles in the embryos, fetuses and offspring created and multiplied by such novel assisted

reproductive technologies and in vitro embryo production strategies as ex vivo fertilization of oocytes with the aid of either classic penetration by capacitated spermatozoa [82,83] or intra-cytoplasmic sperm injection [84,85] and somatic cel cloning [86,87].

The literature states that the cellular mechanisms in oocytes caused by high temperatures are not fully explored. However, we believe that the evidence that some mechanisms occurring during heat stress are related to the increase in reactive oxygen species supports the use of antioxidants for the thermal protection of oocytes during the in vitro maturation process. We also consider it particularly important that IGF1 has an effect on the increased survival of embryos exposed to heat stress during the in vitro process. Undoubtedly, an interesting issue is the positive effect of HSP70 on the developmental competence of oocytes subjected to heat stress and the resulting clear improvement in their quality during in vitro production. A very important aspect of improving the quality of oocytes and embryos in vivo is the cooling of cows during hot days. This is a very important issue to protect the health of dairy cattle, and the developing pregnancy and contributes to improving the profitability of the farm.

REVIEWER: “For example, add some flow charts of heat stress-related mechanisms. This will enable more in-depth discovery.”

Answer: A new figure showing heat stress-related mechanisms has been created and added respectively. Please note Figure 4 in the main MS.

Reviewer 2 Report

GENERAL COMMENTS: the authors showed an important review update regarding the effects of heat stress on the reproduction of lactating cows, mainly on the oocytes and early embryonic development. This is a trending topic, as exposed in previous systematic reviews. Thus, the compilation of the recent findings in this area is essential for scientists aiming to develop new studies regarding epigenetics, reproduction and embryonic developments in dairy production under climate emergency scenarios. The review is well-written, easy to follow, and has well-elaborated figures and schemes. I did not find any substantial mistake that impaired my understanding of the topic. I suggest doing an in-depth English review regarding grammar. Minor errors are observed throughout the text. After this review, I recommend the acceptance of this manuscript as presented.

Author Response

MS ID: cells-2073278

Dear Academic Editor,

Dear Reviewers,

Thank you for inviting us to respond to the very thoughtful and constructive reviewer comments. We greatly appreciate the Editor’s and Reviewers’time and believe our revised manuscript has become more well-rounded as a result.

We have incorporated all suggestions throughout the manuscript, which are highlighted in red in the attached file. Below is a point-by-point response to reviewers’ comments to clarify which edits were made.

We are happy to respond to additional requests if they arise.

Sincerely,

Chandra S. Pareek

Please note our following explanations:

Detailed answers to Reviewer 2:

REVIEWER: “I suggest doing an in-depth English review regarding grammar. Minor errors are observed throughout the text. After this review, I recommend the acceptance of this manuscript as presented.”

Answer: The English has been checked with Grammarly and by a Native speaker.

Reviewer 3 Report

28th November, 2022

Review of Manuscript ID: cells-2073278, by K. Miętkiewska et al., entitled “Effects of heat stress on bovine oocytes and early embryonic development - an update that is intended for publication in Cells

(The Microsoft Word file as Reviewer Attachment for Manuscript ID cells-2073278 Cells 28th November 2022 has also been added)

The Authors of the present study provide comprehensive data about the impacts of thermal stressor on not only the developmental competences, but also intracellular and molecular quality parameters of oocytes, embryos, fetuses, neonates and adult specimens in cattle (Bos taurus).

The paper is very interesting and relatively well written in English. The Authors comment extensively on the threat triggered by global warming (permanent hyperthermy) in the context of its negative effects on the fertility and fecundity in cattle. The text is enriched by  carefully designed Figures well illustrating the contents discussed. However, in my opinion, the following points should be considered prior to the acceptance of manuscript for publication as has been detailed below:

1) Relevant affiliation has to be assigned to Prof. Dr. Chandra S. Pareek.

2)  The paragraphs included between the lines 13 and 17 of the Abstract section:

Cows from calves exposed to thermal stress in utero have lower milk yield in their lifetime, exhibit immunological disorders, have lower birth weight and live shorter lives. In order to protect cow reproduction, should be taken in to account the effects of heat stress at the cellular and molecular level and be familiar with the methods of mitigating its effects.

have to be re-edited to the following sentences:

Cows from calves exposed to thermal stress in utero have lower milk yield in their lifetime, exhibit immunological disorders, have lower birth weight and display a shorter lifespan related to the expedited aging. In order to protect cow reproduction, the effects of heat stress at the intracellular and molecular levels should be tracked step by step and the impacts of dysregulation of thermal homeostasis (i.e., hyperthermy) should be taken into account.

3)  The order and extent of the keywords has to be re-edited as follows:

Keywords: cattle, heat stress, heat shock, oocyte, cumulus cell, granulosa cell, embryo, blastocyst, epigenetics, oxidative stress, mammary gland

4) The sentence included between the lines 124 and 126 within the Subsection 3. The influence of heat stress on granulosa cells:

The studies conducted by [9] showed that the relative amount of……..

has to re-edited to the following form:

The studies conducted by Rispoli et al. [9] showed that the relative amount of…….

5)  The descriptions within the Figure 2 (on the page 4)

Chnages in transcript levels of:

deminished cumulus cell expansion

have to be corrected to the following forms:

Changes in transcript levels of:

Diminished cumulus cell expansion

6) The Conclusions sections is deprived of important future goals and research directions resulting from the investigations carried out by the Authors. Therefore, the following paragraphs and in-text citations of 6 detailed References have to be added by the Authors at the end of the Conclusions section (between the lines 334 and 335) as shown below:

Finally, precisely recognizing the biological factors responsible for heat stress-mediated induction of impairments in reproductive physiology and endocrinology in cattle and other mammalian species might bring about the multifaceted estimation of genomic, epigenomic, transcriptomic and proteomic profiles in the embryos, fetuses and offspring created and multiplied by such novel assisted reproductive technologies and in vitro embryo production strategies as ex vivo fertilization of oocytes with the aid of either classic penetration by capacitated spermatozoa [82,83] or intraooplasmic sperm microinjection [84,85] and somatic cell cloning [86,87].

[82] Sprícigo, J.F.W.; Guimarães, A.L.S.; Cunha, A.T.M.; Leme, L.O.; Carneiro, M.C.; Franco, M.M.; Dode, M.A.N. Using Cumulus Cell Biopsy as a Non-Invasive Tool to Access the Quality of Bovine Oocytes: How Informative Are They? Animals 2022, 12, 3113. doi: 10.3390/ani12223113.

[83] Anzalone, D.A.; Palazzese, L.; Czernik, M.; Sabatucci, A.; Valbonetti, L.; Capra, E.; Loi, P. Controlled spermatozoa-oocyte interaction improves embryo quality in sheep. Sci. Rep. 2021, 11, 22629. doi: 10.1038/s41598-021-02000-z.

[84] Gorczyca, G.; Wartalski, K.; Romek, M.; Samiec, M.; Duda, M. The Molecular Quality and Mitochondrial Activity of Porcine Cumulus-Oocyte Complexes Are Affected by Their Exposure to Three Endocrine-Active Compounds under 3D In Vitro Maturation Conditions. Int. J. Mol. Sci. 2022, 23, 4572. doi: 10.3390/ijms23094572.

[85] Hochi, S.; Ide, M.; Ueno, S.; Hirabayashi, M. High survival of bovine mature oocytes after nylon mesh vitrification, as assessed by intracytoplasmic sperm injection. J. Reprod. Dev. 2022, 68, 335–339. doi: 10.1262/jrd.2022-053.

[86] Samiec, M.; Skrzyszowska, M. Extranuclear Inheritance of Mitochondrial Genome and Epigenetic Reprogrammability of Chromosomal Telomeres in Somatic Cell Cloning of Mammals. Int. J. Mol. Sci. 2021, 22, 3099. doi: 10.3390/ijms22063099.

[87] Yu, T.; Meng, R.; Song, W.; Sun, H.; An, Q.; Zhang, C.; Zhang, Y.; Su, J. ZFP57 regulates DNA methylation of imprinted genes to facilitate embryonic development of somatic cell nuclear transfer embryos in Holstein cows. J. Dairy Sci. 2022, S0022-0302(22)00668-3. doi: 10.3168/jds.2022-22427.

7) The References section has to be prepared according to the requirements of Cells.

In conclusion, the paper can be accepted for publication in Cells, provided that all the minor points will have been taken into account by the Authors as has been above-detailed and precisely recommended by the Reviewer.

Author Response

MS ID: cells-2073278

Dear Academic Editor,

Dear Reviewers,

Thank you for inviting us to respond to the very thoughtful and constructive reviewer comments. We greatly appreciate the Editor’s and Reviewers’time and believe our revised manuscript has become more well-rounded as a result.

We have incorporated all suggestions throughout the manuscript, which are highlighted in red in the attached file. Below is a point-by-point response to reviewers’ comments to clarify which edits were made.

We are happy to respond to additional requests if they arise.

Sincerely,

Chandra S. Pareek

Please note our following explanations:

Detailed answers to Reviewer 3:

REVIEWER: However, in my opinion, the following points should be considered prior to the acceptance of manuscript for publication as has been detailed below:

1) Relevant affiliation has to be assigned to Prof. Dr. Chandra S. Pareek.

Answer: Thank you for the comment regarding the affiliation. We have fixed this issue:

Klaudia Miętkiewska 1 , Pawel Kordowitzki 1 , Chandra Pareek 1,2, *

1 Department of Preclinical and Basic Sciences, Faculty of Biological and Veterinary Sciences, Nicolaus Copernicus University, Torun, Poland

2 Division of Functional Genomics in Biological and Biomedical Research, Interdisciplinary Center for Modern Technologies,

Nicolaus Copernicus University, Torun, Poland.

* Correspondence: [email protected]

REVIEWER: “2)  The paragraphs included between the lines 13 and 17 of the Abstract section:

Cows from calves exposed to thermal stress in utero have lower milk yield in their lifetime, exhibit immunological disorders, have lower birth weight and live shorter lives. In order to protect cow reproduction, should be taken in to account the effects of heat stress at the cellular and molecular level and be familiar with the methods of mitigating its effects.

have to be re-edited to the following sentences:

Cows from calves exposed to thermal stress in utero have lower milk yield in their lifetime, exhibit immunological disorders, have lower birth weight and display a shorter lifespan related to the expedited aging. In order to protect cow reproduction, the effects of heat stress at the intracellular and molecular levels should be tracked step by step and the impacts of dysregulation of thermal homeostasis (i.e., hyperthermy) should be taken into account.”

Answer: Thank you for the valuable information regarding the rewording of the mentioned sentences, and we have addressed this respectively.

REVIEWER: “3)  The order and extent of the keywords has to be re-edited as follows:

Keywords: cattle, heat stress, heat shock, oocyte, cumulus cell, granulosa cell, embryo, blastocyst, epigenetics, oxidative stress, mammary gland.”

Answer: done

REVIEWER: “4) The sentence included between the lines 124 and 126 within the Subsection 3. The influence of heat stress on granulosa cells:

The studies conducted by [9] showed that the relative amount of……..

has to re-edited to the following form:

The studies conducted by Rispoli et al. [9] showed that the relative amount of…….”

Answer: done

REVIEWER: “5)  The descriptions within the Figure 2 (on the page 4)

Changes in transcript levels of:

diminished cumulus cell expansion

have to be corrected to the following forms:

Changes in transcript levels of:

Diminished cumulus cell expansion”

Answer: done.

REVIEWER: “6) The Conclusions section is deprived of important future goals and research directions resulting from the investigations carried out by the Authors. Therefore, the following paragraphs and in-text citations of 6 detailed References have to be added by the Authors at the end of the Conclusions section (between lines 334 and 335) as shown below:

Finally, precisely recognizing the biological factors responsible for heat stress-mediated induction of impairments in reproductive physiology and endocrinology in cattle and other mammalian species might bring about the multifaceted estimation of genomic, epigenomic, transcriptomic and proteomic profiles in the embryos, fetuses and offspring created and multiplied by such novel assisted reproductive technologies and in vitro embryo production strategies as ex vivo fertilization of oocytes with the aid of either classic penetration by capacitated spermatozoa [82,83] or intra cytoplasmic sperm injection [84,85] and somatic cell cloning [86,87].

Answer: Thank you for your suggestion to include this valuable information in the conclusion of our review paper. We would like to inform you that we have placed the text you provided in the designated place, and the new reference order is follows:

[82] Sprícigo, J.F.W.; Guimarães, A.L.S.; Cunha, A.T.M.; Leme, L.O.; Carneiro, M.C.; Franco, M.M.; Dode, M.A.N. Using Cumulus Cell Biopsy as a Non-Invasive Tool to Access the Quality of Bovine Oocytes: How Informative Are They? Animals 2022, 12, 3113. doi: 10.3390/ani12223113.

[83] Anzalone, D.A.; Palazzese, L.; Czernik, M.; Sabatucci, A.; Valbonetti, L.; Capra, E.; Loi, P. Controlled spermatozoa-oocyte interaction improves embryo quality in sheep. Sci. Rep. 2021, 11, 22629. doi: 10.1038/s41598-021-02000-z.

[84] Gorczyca, G.; Wartalski, K.; Romek, M.; Samiec, M.; Duda, M. The Molecular Quality and Mitochondrial Activity of Porcine Cumulus-Oocyte Complexes Are Affected by Their Exposure to Three Endocrine-Active Compounds under 3D In Vitro Maturation Conditions. Int. J. Mol. Sci. 2022, 23, 4572. doi: 10.3390/ijms23094572.

[85] Hochi, S.; Ide, M.; Ueno, S.; Hirabayashi, M. High survival of bovine mature oocytes after nylon mesh vitrification, as assessed by intracytoplasmic sperm injection. J. Reprod. Dev. 2022, 68, 335–339. doi: 10.1262/jrd.2022-053.

[86] Samiec, M.; Skrzyszowska, M. Extranuclear Inheritance of Mitochondrial Genome and Epigenetic Reprogrammability of Chromosomal Telomeres in Somatic Cell Cloning of Mammals. Int. J. Mol. Sci. 2021, 22, 3099. doi: 10.3390/ijms22063099.

[87] Yu, T.; Meng, R.; Song, W.; Sun, H.; An, Q.; Zhang, C.; Zhang, Y.; Su, J. ZFP57 regulates DNA methylation of imprinted genes to facilitate embryonic development of somatic cell nuclear transfer embryos in Holstein cows. J. Dairy Sci. 2022, S0022-0302(22)00668-3. doi: 10.3168/jds.2022-22427.

7) The References section has to be prepared according to the requirements of Cells.

7) REVIEWER: “7) The References section has to be prepared according to the requirements of Cells.

Answer: done

COMMENT of the ACADEMIC EDITOR:

Academic Editor: „We have received the reports from three expert Reviewers on your manuscript, "Effects of heat stress on bovine oocytes and early embryonic development- an update", which you submitted to Cells. Based on the advice received, I have decided that your manuscript could be considered for publication provided you incorporate the Reviewer's comments and suggestions. In particular, you should take into account the concerns raised by Reviewer 1 trying to treat more clearly and in detail the cellular and molecular dysfunctions induced by HS on folliculogenesis, oocytes/embryos, uterus etc. The Authors should improve the manuscript by describing more in depth those issues and providing a personal interpretation of literature data and strategies to mitigate the effects of HS in vitro and to prevent HS in vivo. „

Answer: Dear Academic Editor Roberto Gualtieri, thank you very much for the opportunity to correct our mistakes. We have done our best to meet the requirements of the reviewers. As suggested by the first reviewer, we have thoroughly explained the mechanism of heat stress in cattle and added diagrams showing changes in cows under the influence of heat stress. We also took into account all the comments of other reviewers. Referring to your observation regarding our interpretation of the effects of heat stress mitigation in vitro and in vivo, we decided to add the following information to the conclusions:

The literature states that the cellular mechanisms in oocytes caused by high temperatures are not fully explored. However, we believe that the evidence that some mechanisms occurring during heat stress are related to the increase in reactive oxygen species supports the use of antioxidants for the thermal protection of oocytes during the in vitro maturation process. We also consider it particularly important that IGF1 has an effect on the increased survival of embryos exposed to heat stress during the in vitro process. Undoubtedly, an interesting issue is the positive effect of HSP70 on the developmental competence of oocytes subjected to heat stress and the resulting clear improvement in their quality during in vitro production. A very important aspect of improving the quality of oocytes and embryos in vivo is the cooling of cows during hot days. This is a very important issue to protect the health of dairy cattle, and the developing pregnancy and contributes to improving the profitability of the farm.

Round 2

Reviewer 1 Report

The author has done a lot of revision work to make the article look more in-depth. But I think these changes are not enough to meet the requirements of the journal. The quality of this review article is average, and there is no good depth. The author should focus on the mechanism route and how to further explore effective information.